# TOWARDS UNDERSTANDING MASKED DISTILLATION

## ABSTRACT

In the realm of self-supervised learning, Masked Image Modeling (MIM) serves as a viable approach for mitigating the dependency on large-scale annotated data, while demonstrating efficacy across a broad spectrum of downstream tasks. A recent variant of MIM known as Masked Distillation (MD) has emerged, which utilizes semantic features instead of low-level features as the supervision. Although prior work has demonstrated its effectiveness in various downstream tasks, the underlying mechanisms for its performance improvements remain unclear. Our investigation reveals that Masked Distillation mitigates multiple forms of overfitting present in the original models, including but not limited to attention homogenization and the representation folding of high layers. Further, we uncover that Masked Distillation introduces beneficial inductive biases stemming from MIM, which are believed to contribute positively to model performance. We also analyze the nuances of the model architecture design and decision-making tendencies in Masked Distillation, revealing inconsistencies with previous research findings.

## 1 INTRODUCTION

Self-supervised Learning (SSL) in computer vision can learn powerful representations from large-scale unlabeled images without relying on human annotations. As a particular instance of SSL, Masked Image Modeling (MIM) (Bao et al., 2021; He et al., 2022) involves occluding regions within the image space and subsequently reconstructing the masked signals. In stark contrast to contrastive learning, which focuses on the consistency of global views, MIM takes its cues from Masked Language Modeling (MLM) in the field of Natural Language Processing (NLP) (Gui et al., 2023). However, a recent line of works has emerged that employs features from vision-language models, e.g. CLIP (Radford et al., 2021), as the supervision, a concept we refer to as Masked Distillation (MD) (Fang et al., 2023; Peng et al., 2022; Hou et al., 2022; Peng et al., 2023). Detailed definitions are provided in Section 2.

Previous works (Peng et al., 2022; 2023) have demonstrated performance improvements through Masked Distillation, as shown in Figure 3c. Models pre-trained with it have been extensively utilized in a wide array of downstream tasks (Fang et al., 2023). However, there is an absence of an explanation regarding the source of performance benefits. We are interested in exploring the underlying mechanisms and aim to elucidate the functioning of Masked Distillation. Preliminary to this, we offer an intuitive understanding of Masked Distillation.

MIM takes a degraded image and attempts to predict the missing information, as in Figure 1c. Researchers have proposed various prediction targets include raw pixels (He et al., 2022), visual tokens (Bao et al., 2021), Histogram of Oriented Gradients (HOG) (Wei et al., 2022a), etc. Although MIM is inspired by MLM in NLP, there exists a fundamental difference between the two modalities. Unlike the semantics in NLP, which naturally emerge from statistical analysis of word frequencies, visual signals are continuous and organic, whereas natural language is discrete and information-rich (Zhou et al., 2021). This leads to a unique challenge: the semantics of visual data are not readily amenable to summarization or compression. Consequently, the inherent focus of MIM on pixel-level reconstruction has been criticized for squandering modeling capacity on reconstructing low-level pixel features (Liu et al., 2023; Chen et al., 2023). Note that visual tokens and HOG features are also merely types of low-level features; therefore, they share characteristics with algorithms based on raw pixel reconstruction (Gui et al., 2023). Previous work has addressed this issue through various indirect approaches, including the MAE's decoder (He et al., 2022) and BEiT's discrete visual tokens

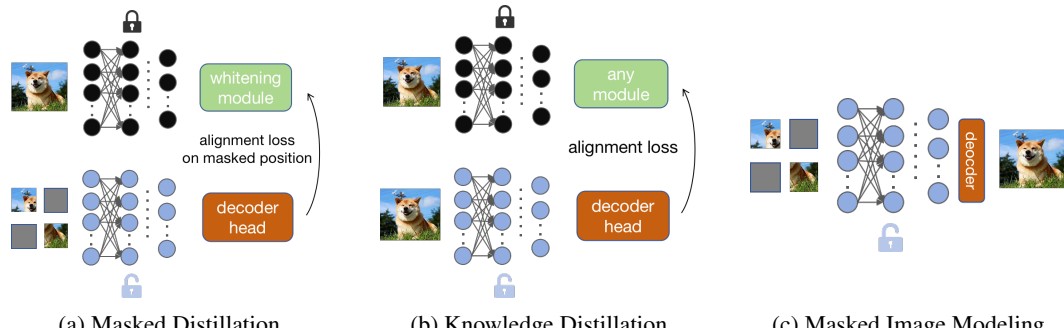

(a) Masked Distillation (b) Knowledge Distillation (c) Masked Image Modeling

Figure 1: (a) Overview of Masked Distillation. A portion of the input signal is corrupted and the model endeavors to reconstruct the corrupted signals as features generated by a whitened teacher model. (b) Overview of Knowledge Distillation. Both the teacher model and the student model are fed the same input, and the model aims to align the features between the two. (c) Overview of Masked Image Modeling. A portion of the input signal is corrupted and the objective of reconstructing is corresponding low-level features, e.g. raw pixels. Note that the connotations of these frameworks extend beyond the essential features presented here for the sake of clarity and conciseness.

(Bao et al., 2021). Employing semantic features as reconstruction targets other than low-level pixels intuitively offers the potential for more advanced and semantically rich feature representations.

In addition to the challenge of incompressible visual semantics, another issue lies in the gap between pre-training and downstream tasks. Research in contrastive learning reveals that the choice of appropriate data augmentation—or degradation methods in the context of MIM—depends on the specific downstream tasks (Tian et al., 2020). Moreover, the literature indicates that naive contrastive learning is ill-suited for dense prediction tasks, while MIM shows superior performance in such scenarios (Xie et al., 2021; He et al., 2022). These observations underscore the existing gap between the pre-training tasks and the downstream tasks. In Section 3, we demonstrate how Masked Distillation endows the pre-trained model with attributes of MIM, which mitigates the discrepancy between the pre-training and downstream tasks. To a certain extent, Masked Distillation facilitates the decoupling of pre-training tasks from downstream tasks, allowing us to focus more on designing powerful pre-training tasks without being overly concerned with the specific requirements of downstream applications.

In this paper, we explore Masked Distillation through a series of analyses. Although former research has traditionally relied on vision-language contrastive learning teachers, such as CLIP, for semantic feature extraction. We extend our inquiry to both supervised and self-supervised learning paradigms to offer a comprehensive perspective.

The contributions of our work are summarized as follows.

- We introduce an analytical framework for the comprehensive understanding of Masked Distillation, with potential applicability extending to other investigations.
- Our analysis reveals that Masked Distillation endows the model with properties akin to MIM, mitigating various forms of overfitting, including attention homogenization and representation folding. We also find that previous research on the relationship between Masked Distillation and decision propensity is insufficiently comprehensive.
- We demonstrate that Masked Distillation results in smoother convergence points. Inspired by our observations, we introduce a simple yet more powerful baseline and provide insights into the model architecture design.

## 2 PRELIMINARIES

In this section, we first introduce Masked Distillation. The key difference between classical Masked Image Modeling and Masked Distillation lies in the choice of the supervision signal. Rather than

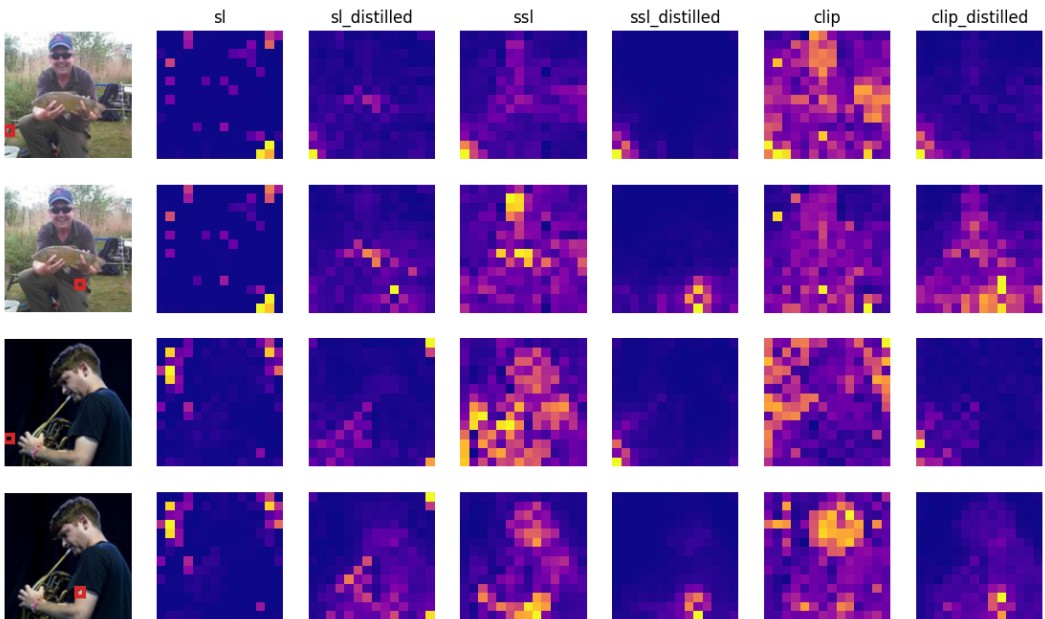

Figure 2: Visualizations of the final attention block. Queries marked with red boxes are displayed in the leftmost column of the image. The figure reveals a pronounced occurrence of attention homogenization in pre-trained models. Masked Distillation ameliorates attention collapse, directing attention toward more relevant and localized regions.

utilizing low-level features, Masked Distillation employs semantic features for supervision. To facilitate a concise and focused discussion, we introduce a minimalist framework.

A typical Masked Distillation framework generally comprises a teacher model $\mathcal{T}$, a whitening module $\mathcal{N}$, a student model $\mathcal{S}$ awaiting training, a decoder head $\mathcal{D}$, a mask generator $\mathcal{M}$, and an objective function $\mathcal{R}$. Refer to Figure 1a for the framework diagram. For a certain input image $\mathcal{I}$, the training task is formulated as

$$\mathcal{L} = \mathcal{M}\big(\mathcal{R}\big(\mathcal{N}(\mathcal{T}(\mathcal{I})), \mathcal{D}(\mathcal{S}(\mathcal{M}(\mathcal{I})))\big)\big), \tag{1}$$

where inner $\mathcal{M}$ serves to corrupt the input signal, while the outer $\mathcal{M}$ is designed for calculating the loss specifically at the corrupted positions. The whitening module $\mathcal{N}$ is crucial to whiten the features from the teacher model, addressing the issue of varying statistical properties across features of different teacher models. The whitening operation contributes to stabilizing the training process and obviating the need for laborious hyperparameter tuning (Saha et al., 2022). The decoder head $\mathcal{P}$ serves to project the features of the student model into the whitened feature space. Notably, the dimensionalities of the teacher and student can be incongruent to accommodate a diverse range of models. The decoder head can be implemented either as a transformer (Hou et al., 2022) or as a simple MLP (Wei et al., 2022b). The other modules are largely consistent with classical MIM (Bao et al., 2021; He et al., 2022). Note that the concept of Masked Distillation is not limited to the Vision Transformer (ViT) (Dosovitskiy et al., 2020), though for the sake of conciseness, ViTs are primarily considered in our discussion.

## 3 MASKED DISTILLATION INHERITS THE PROPERTIES OF MASKED IMAGE MODELING

Masked Distillation is a specialized form of MIM, where the reconstruction target is semantic features rather than normalized original pixels (He et al., 2022) or visual tokens (Peng et al., 2022). Therefore, a question arises: What differentiates Masked Distillation from MIM? We propose a natural hypothesis: Masked Distillation imbues the model with attributes inherent to MIM.

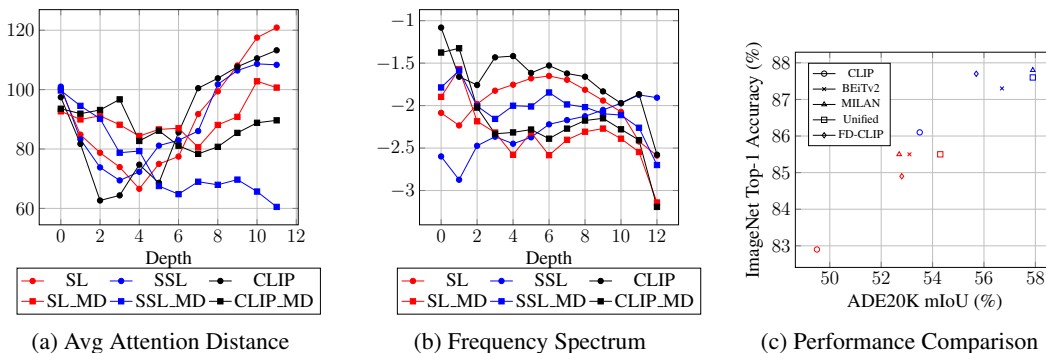

(a) Avg Attention Distance      (b) Frequency Spectrum      (c) Performance Comparison

Figure 3: (a) Average attention distance. Masked Distillation reduces the average attention distance compared to the original model. (b) Relative log amplitude spectrum of frequencies. Masked Distillation amplifies the model's preference for low-frequency features, while enhancing the preference for high-frequency features in the lower layers. (c) Performance comparison across various algorithms, specifically including CLIP (Radford et al., 2021), BEITv2 (Peng et al., 2022), MILAN (Hou et al., 2022), Unified (Peng et al., 2023), and FD-CLIP (Wei et al., 2022b). Red represents ViT-B, and blue represents ViT-L.

In this section, we validate the hypothesis. First, we identify the properties of MIM models reported in prior literature: attention locality, diversity, and specific frequency biases (Xie et al., 2023a; Park et al., 2022). We extend the sources of semantic features to three representative models: DeiT (supervised model) (Touvron et al., 2021), DINO (self-supervised contrastive model) (Caron et al., 2021), and CLIP (vision-language contrastive learning model) (Radford et al., 2021), standardizing the architecture to ViT-B/16. Except for CLIP, which is pre-trained on private data, the other models are pre-trained on ImageNet-1k (Russakovsky et al., 2015). We employ officially released models and train them with the unified codebase (Peng et al., 2022). We have included the hyperparameter settings in Appendix D.

**Attention homogenization exists in pretrained models.** In Figure 2, we visualize the attention maps across different models. Initially, we observe that two contrastive learning models exhibit attention homogenization—namely, the model's response occurs across a broad and unrelated area for specific queries and area of responses are identical for different queries. This phenomenon, reported in Park et al. (2022) is termed as attention homogenization.

In supervised models, the response regions are consistently similar and tend to be oriented towards the background areas, distinguishing it from DINO and CLIP. This is attributed to supervised learning's focus on task-relevant regions, with other information being disregarded. Interestingly, ViTs trained with supervised learning exhibit a high response to the background, which diverges from traditional understanding where the response locations should be correlated with the query (Dosovitskiy et al., 2020). In Figure 10, we also present a visualization of the distribution of attention entropy across different heads for various models. Pre-trained models tend to exhibit homogeneous and elevated attention entropy at higher layers, indicating that different heads focus on similar information. The Masked Distillation model effectively diversifies the attention across different heads.

**Masked Distillation inherits the inductive bias of attention from MIM.** Xie et al. (2023a) demonstrates that MIM possesses diversified and more localized attention compared to supervised and contrastive self-supervised learning paradigms. In Figure 2, Masked Distillation alleviates this issue by localizing the model's attention focus around the query. It also diversifies the attention, as reported in Figure 10. Intriguingly, the locality and diversity in models distilled from the supervised model is weaker compared to other models. This phenomenon may be attributed to their extreme homogenization of background responses, suggesting that the degree of locality and diversity attained is contingent upon the original model.

To quantify locality, we calculated the average attention distance for different models. Average attention distance, i.e. the average distance of the model's attention, is the integral of the model's attention with respect to token distance. The average attention across models is presented in Figure

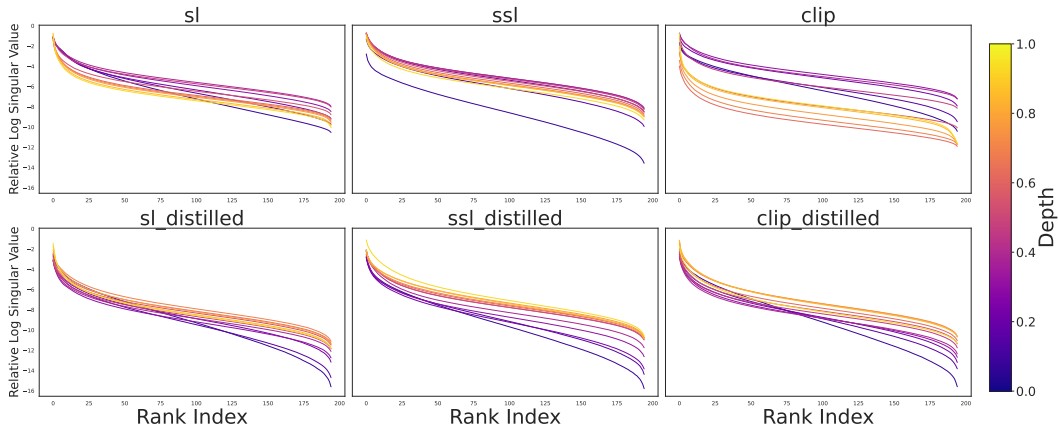

Figure 4: The visualization of token-level relative singular value spectrum with lighter color means deeper layer. The phenomenon of representation folding is observed in pre-trained models, where features of high layers are compressed into a reduced space. Masked Distillation progressively enhances the discriminative capability of tokens by increasing the token-level relative singular values during the model forward process.

3a. It shows that supervised models exhibit the highest average attention distance, consistent with the attention visualization. For all models, Masked Distillation reduces their average attention distance, verifying an enhanced locality in the attention module. Noteworthily, the average attention in the lower layers of the model increased. Given that distillation occurs at the higher layers of the model, this phenomenon does not contradict our understanding of Masked Distillation.

**Masked Distillation shifts the frequency bias.** Park et al. (2022) reveals that MIM exhibits a preference for low-frequency features in the higher layers, while favoring high-frequency features in the lower layers. Contrastive Predictive Coding (CPC) (van den Oord et al., 2019) posits that contrastive learning seeks invariant patterns within input signals. Therefore, contrastive learning can be considered as having a preference for low-frequency information. In light of this, we are interested in exploring how Masked Distillation shifts frequency preferences. Specifically, we performed Fourier transform on the features at each layer and plotted the relative log amplitude spectrum in Figure 3b. The relative log amplitude refers to the amplitude difference between the highest and lowest frequencies.

Masked Distillation consistently enhances the model's preference for low-frequency components. Surprisingly, a heightened preference for high-frequency elements is observed in the lower layers of the models. This phenomenon occurs because the higher layers of the MIM model implicitly act as a decoder, which inherently exhibits a low-frequency bias. This, in turn, facilitates the lower layers' utilization of high-frequency features (Park et al., 2022).

**Conclusion.** In this section, we have demonstrated that Masked Distillation imbues models with key attributes of MIM, including the specific frequency bias, locality, and diversity. These attributes of MIM have been found to generalize well to various downstream tasks (He et al., 2022; Park et al., 2022; Xie et al., 2023a).

## 4 HOW DOES MASKED DISTILLATION TRANSFER WELL IN DOWNSTREAM TASKS?

To further investigate the origins of the performance gains attributed to Masked Distillation, our analysis pivots on the dual perspectives of representation capacity and model optimization. The model involved remains consistent with the one described in Section 3.

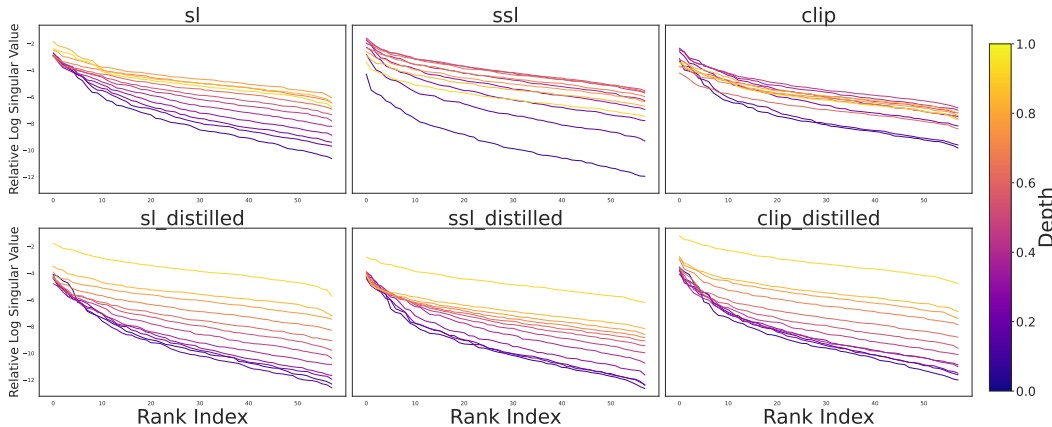

Figure 5: The visualization of image-level relative singular value spectrum with lighter color means deeper layer. In consideration of capacity constraints, we have truncated the indices. Masked Distillation significantly increases the distribution volume of high layers, thereby enhancing the discriminative capability of images.

**Masked Distillation enhances the expressive capability of the attention module.** In Section 3, we highlighted how Masked Distillation enables the attention module to mimic the behavior of MIM. In this section, we further demonstrate that Masked Distillation significantly amplifies the representation capacity of the attention module.

To quantify this capability, we introduce Normalized Mutual Information (NMI). A low NMI indicates that the query tokens and key tokens are not particularly related, whereas a high NMI signifies that the similarity between query tokens and key tokens carries rich information. Specifically, the definition of NMI is $\frac{I(q,k)}{\sqrt{H(q)H(k)}}$ where $I(\cdot, \cdot)$ denotes the mutual information and $H(\cdot)$ is the marginal entropy. In the case of high-dimensional data or large datasets, computing NMI is computationally infeasible. Given the nature of the self-attention, it is natural to take attention distribution under a particular query $q$ as $p(k|q)$, while assuming a uniform distribution for the queries.

Figure 6a illustrates the NMI values across different layers of models. In their un-distilled states, models manifest exceedingly low NMI values, indicating that their attention focuses on similar patterns, which is not conducive to effective task transfer. In line with visual observation, the SSL model registers the lowest NMI values. The incorporation of Masked Distillation consistently enhances the NMI performance. Notably, a slight decrease in NMI is observed in lower layers, which resonates with our findings on average attention distance. This inconsistent behavior between higher and lower layers manifests across multiple dimensionalities and is further explored in the context of representation distribution in Section 4.

**Masked Distillation mitigates the representation folding of high layers.** Given that the expressive capability of the self-attention module has been enhanced and performance improvements in downstream tasks has been achieved, we hypothesize that the model is capable of extracting more powerful features. To validate our conjectures, we present the token-level and image-level relative singular value spectrum of representations in Figure 4 and Figure 5, respectively. The relative singular value spectrum is obtained by Singular Value Decomposition (SVD) on the representation which reflects the effective volume of distributions in the representation space (Park et al., 2022). Larger singular values indicate greater volume in the representation space. The relative log singular value ($\Delta$ Log singular value) is the difference with the highest log singular values.

Figure 4 presents the relative singular value spectrum of token-level features, reflecting how the model transforms tokens. Specifically, we calculate the relative singular values of tokens in a single image and average them over the ImageNet validation set. In the context of the ViT, we are particularly interested in investigating how the model learns the representation of tokens. The higher layers of pre-trained models exhibit diminished relative singular values in comparison to the lower

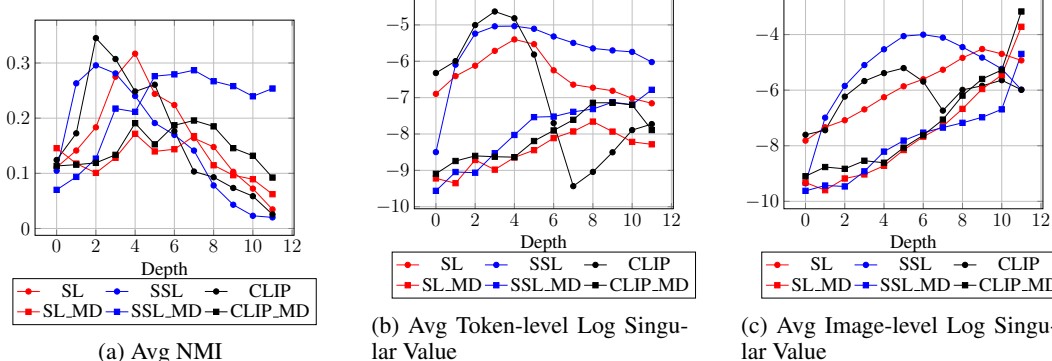

(a) Avg NMI

(b) Avg Token-level Log Singular Value

(c) Avg Image-level Log Singular Value

Figure 6: (a) Average NMI values. Pre-trained models exhibit extremely low NMI at higher layers, an issue mitigated by Masked Distillation. (b) & (c) Average token-level log singular value spectrum and average image-level log singular value spectrum.

or intermediate layers, meaning the representations of high layers are folding into a low-dimensional space. This phenomenon can be explained through Information Bottleneck Theory (Tishby & Zaslavsky, 2015) as the higher layers extract task-specific features, leading the higher layers to overfit the pre-training task and can hardly distinguish tokens. Masked Distillation markedly ameliorates the phenomenon of high-level representation folding. This may facilitate the model's transferability to downstream tasks, rather than overfitting to the pre-training tasks. We also visualize the image-level relative singular value spectrum in Figure 5. Our findings reveal that analogous observations hold true for image-level representations. To demonstrate the significance of high layers during transferring, we present the relationship between accuracy and the number of frozen layers in Figure 8b. The results reveal that fine-tuning merely the top layers suffices to approach full accuracy, suggesting that mitigating overfitting in higher layers could be pivotal for successful transfer.

To offer a more quantitative perspective, Figure 6b and Figure 6c display the average relative singular value spectrum on different layers. Notably, we observe a pronounced phenomenon of representation folding in CLIP, characterized by a sharp decline in representation capacity at higher layers. This suggests that vision-language models like CLIP may process features in a distinctive manner compared to others.

**Masked Distillation smooths the loss landscape.** As above mentioned, we observed the phenomenon of representation folding, especially the CLIP model. We identify that this behavior hinders the model optimization in downstream tasks. To quantify the nature of the optimization process, we computed the Hessian matrix of the model.

A neural network may involve millions of parameters, and calculating second-order statistics requires computational resources that are orders of magnitude greater than those for first-order statistics. In order to accelerate the computation, we employ the Power Iteration Method (Yao et al., 2020). For a given random vector $v$, we compute the application of the Hessian to $v$:

$$Hv = \frac{\partial g_\theta^\mathrm{T}}{\partial \theta} v = \frac{\partial g_\theta^\mathrm{T}}{\partial \theta} v + g_\theta^\mathrm{T} \frac{\partial v}{\partial \theta} = \frac{\partial g_\theta^\mathrm{T} v}{\partial \theta}. \tag{2}$$

With the Power Iteration Method, we can observe the Hessian spectrum without explicitly computing the Hessian matrix. In terms of computational expense, the calculation of the final term is equivalent to computing first-order statistics. We computed the top-k (2 for default) eigenvalues and eigenvectors of the Hessian matrix for finetuned CLIP, DINO, and their respective Masked Distillation counterparts on the ImageNet validation set. We deliberately excluded the supervised model from our analysis, as it has been trained on ImageNet with labeled data and are susceptible to overfitting. Firstly, We visualize the loss landscape in Figure 7. It reveals that the distilled models exhibit a smoother loss landscape at its convergence point.

Sharpness, i.e. the largest eigenvalue, of the Hessian matrix influences the optimization stability (Fu et al., 2023). We visualize the absolute sharpness during the finetuning process in Figure 7c. Firstly, we observe that both the CLIP model and its Masked Distillation counterpart exhibit signif-

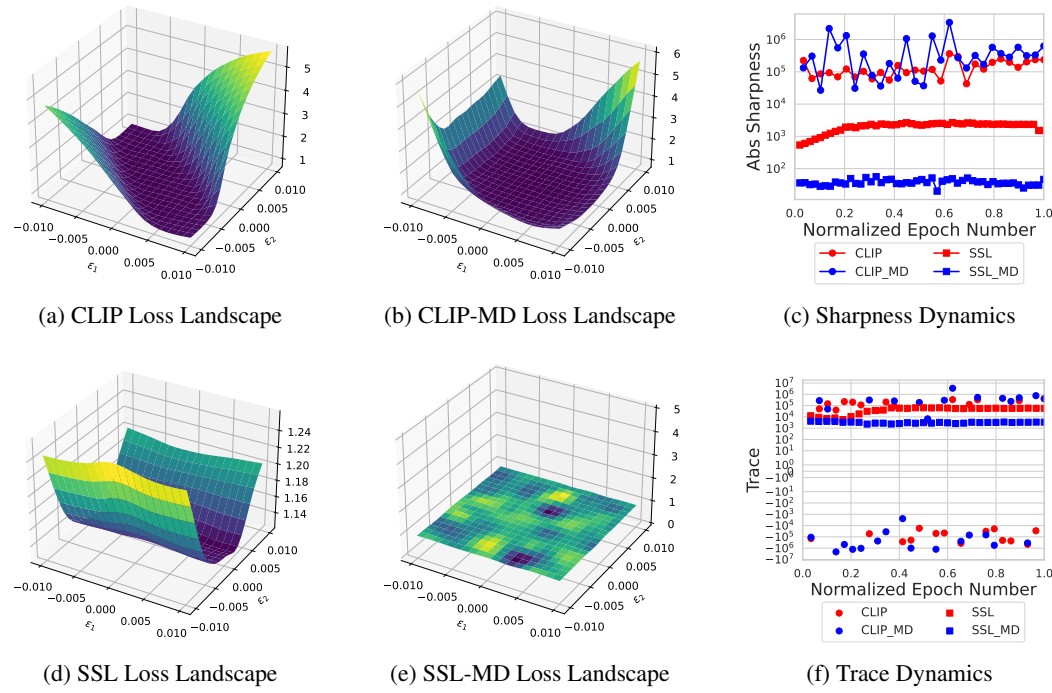

(a) CLIP Loss Landscape      (b) CLIP-MD Loss Landscape      (c) Sharpness Dynamics

(d) SSL Loss Landscape      (e) SSL-MD Loss Landscape      (f) Trace Dynamics

Figure 7: The parametric loss landscapes of original and Masked Distillation models on the ImageNet validation set are presented in Figure 7a,7b,7d,7e by perturbing the model parameters at the end of training across the first and second Hessian eigenvector. It reveals that the point to which the distilled models converge at a smoother point. In Figures 7c and 7f, the variations in absolute sharpness and trace of the entire network during the training process are presented respectively.

icantly high absolute sharpness, indicating the difficulty of optimization. This observation leads us to explore the impact of learning rates on training in Figure 8b, as the proper learning rates are potentially constrained by sharpness (Cohen et al., 2020). Surprisingly, we find that the original CLIP model, after properly reducing the learning rate, demonstrates a higher fine-tuning accuracy: 84.8% as opposed to 82.9% (Wei et al., 2022b). This insight guides us toward introducing a better baseline, achievable through the straightforward adjustment of reducing the learning rate. Training hyperparameters can be found in Appendix D. Moreover, we discover that Masked Distillation renders the model less sensitive to the learning rate. We also observe that Masked Distillation significantly lowers the amplitude of sharpness of the SSL model.

We also examined the variations in trace values. We observed that in the middle and later stages of training, CLIP tends to manifest negative trace values, which can impede model optimization. In contrast, the negative trace distribution in Masked Distillation counterpart is more uniform. Additionally, we noticed a rapid increase in trace values during the mid-phase of DINO's training, whereas the Masked Distillation counterparts remained stable.

## 5   DEEP DIVE INTO MODEL DESIGN AND DECISION PREFERENCES

In prior sections, we have explored the impact of Masked Distillation from various perspectives. In this section, we delve into the intricacies of model design and examine the effects of Masked Distillation on decision preferences.

**Should we employ a tokenizer?** BEiT (Peng et al., 2022) utilizes a tokenizer to extract the visual tokens as supervision for masked tokens, yet EVA (Fang et al., 2023) contends that the use of a tokenizer is not imperative. To elucidate the distinctions between tokenized and direct feature distillation, we employ the aforementioned analytical framework. In alignment with EVA, we adopt CLIP

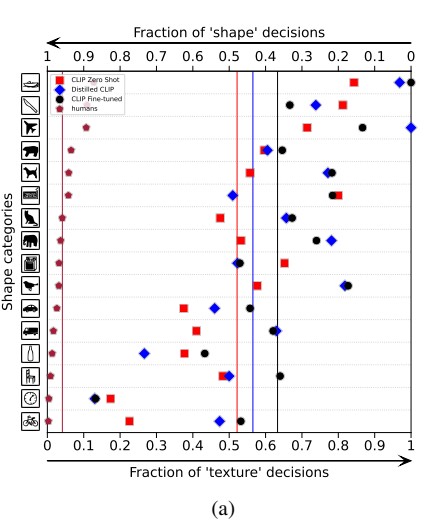

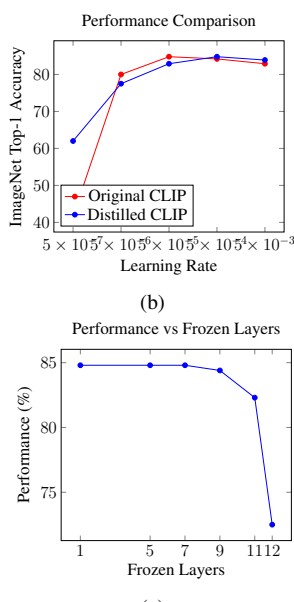

Figure 8: (a) Analysis of shape bias under the teacher supervision in CLIP ViT-B/16. (b) Relationship between the ImageNet top-1 accuracy and the learning rate. (c) Correlation between the accuracy of directly fine-tuned CLIP and the number of frozen transformer blocks. The numbers 1 to 11 indicate the number of frozen transformer blocks, organized in an ascending hierarchical order from lower to higher layers. Number 12 denotes the linear probe.

as the teacher. In Figure 9, we compare the tokenizer against a fully connected layer as the decoder. We observe that employing a simple and shallow decoder brings the model's behavior closer to that of the baseline. Due to the design of discrete tokens, the tokenizer is capable of ignoring irrelevant details, thereby manifesting a stronger preference for low-frequency information. Although both approaches lead to performance gains in downstream tasks, tasks that favor low-frequency information may benefit more from the tokenizer (Peng et al., 2022; Hou et al., 2022; Peng et al., 2023).

**Does Masked Distillation shift the decision preferences?** We are interested in the model's decision preferences. Prior research posits that Masked Distillation enhances the shape bias (Peng et al., 2023). We extend this understanding and find that the original models display the most pronounced shape bias. Fine-tuning pre-trained models tends to bias them towards texture. However, employing Masked Distillation ameliorates this shift. More strictly speaking, Masked Distillation mitigates the texture bias introduced by fine-tuning. In a more significant context, when we extend the test scenarios to 17 diverse out-of-distribution (OOD) datasets from Geirhos et al. (2021), we find that Masked Distillation does not make the model's decisions more human-like. More experiment details can be found in Appendix C.

## 6 CONCLUSION

To facilitate a comprehensive understanding of Masked Distillation, we have conducted a series of analyses. Initially, we introduce Masked Distillation via a streamlined framework before extending the scope of our investigation from the singular CLIP model to three different learning paradigms. Our study reveals that Masked Distillation embodies some of the most salient characteristics of MIM, including locality, diversity, and specific frequency biases. We further identify multiple forms of overfitting present in pre-trained models, such as attention homogenization and representation folding. Our analyses indicate that Masked Distillation effectively mitigates these overfitting issues, guiding the model toward smoother optimization landscapes. Based on our observations, we also propose a simple yet more powerful baseline.

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

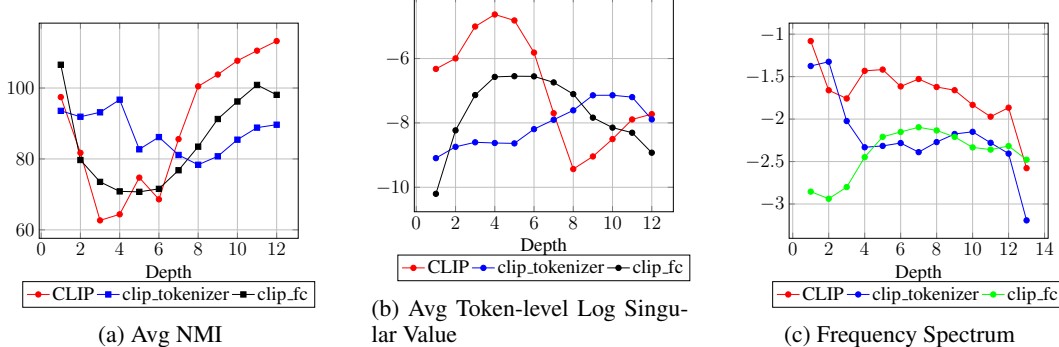

Figure 9: The ablation study concerning the use of tokenizers, suffix *tokenizer* signifies the employment of a tokenizer, whereas suffix *fc* denotes the utilization of a fully connected layer as the decoder.

## A   RELATED WORK

**Masked Image Modeling.**   Unsupervised learning which aims at learning representations from unlabeled data has been a tradition in machine learning. Before SSL, many unsupervised learning methods have been proposed, including probabilistic latent variable models (Sussmann, 1988; Hinton et al., 2006; Salakhutdinov et al., 2007), variants of auto-encoders (Vincent et al., 2008; Kingma & Welling, 2022; Rezende et al., 2014). Recently, SSL has been a very active area of research due to its ability to scale data more effectively than supervised learning by handling unlabeled data (Xie et al., 2023b; Radford et al., 2021). SSL typically involves the design of a pretext task (Gui et al., 2023) and masked image modeling (MIM) is one of the most popular pretext tasks.

MIM functions as a denoising autoencoder (Vincent et al., 2008), with the fundamental idea of accepting degraded input and attempting to predict the missing information. For MIM, the three most essential elements are the degradation method, prediction target, and model architecture. Researchers have proposed various degradation methods, including blockwise mask (Bao et al., 2021), random mask (He et al., 2022), adversarial mask (Shi et al., 2022), frequency mask (Xie et al., 2022), etc. Different prediction targets include raw pixels (He et al., 2022), visual tokens (Bao et al., 2021), Histogram of Oriented Gradients (HOG) (Wei et al., 2022a), etc. Raw pixel prediction typically involves using a decoder to reconstruct pixels from visual features, whereas visual token prediction employs a discrete autoencoder to train visual tokens as the reconstruction target.

**Knowledge Distillation.**   Knowledge distillation involves transferring knowledge from one model, termed the teacher, to another model, known as the student. The student model emulates the behavior of the teacher model by aligning their logits (or intermediate layer activations). A category of research closely related to this paper is feature-level distillation (Heo et al., 2019a;b), referring to approaches that compel the student model to emulate the intermediate-layer feature outputs of the teacher model, even when their dimensionalities are misaligned. Another category of work in knowledge distillation conceptualizes the relationships between input signals as the knowledge to be transferred (Tian et al., 2019; Tung & Mori, 2019). Generally, knowledge distillation focuses on knowledge migration between models, often from larger to smaller architectures. In our study, we concentrate on teacher-student models with identical architectures and place greater emphasis on downstream task performance.

## B   DECODER HEAD DESIGN

To investigate the differences between utilizing and not utilizing a tokenizer, we introduced a control group that employs a fully connected layer as the decoder. Representative results are presented in Figure 9. Overall, the use of a simple fully connected layer brings the model's behavior closer to that of the baseline model. This is in line with the role of the decoder in MIM (He et al., 2022; Bao et al., 2021; Park et al., 2022), which encodes fine-grained details and serves, in part, as a representation learner, thus making a shallow decoder more amenable to preserving the details of

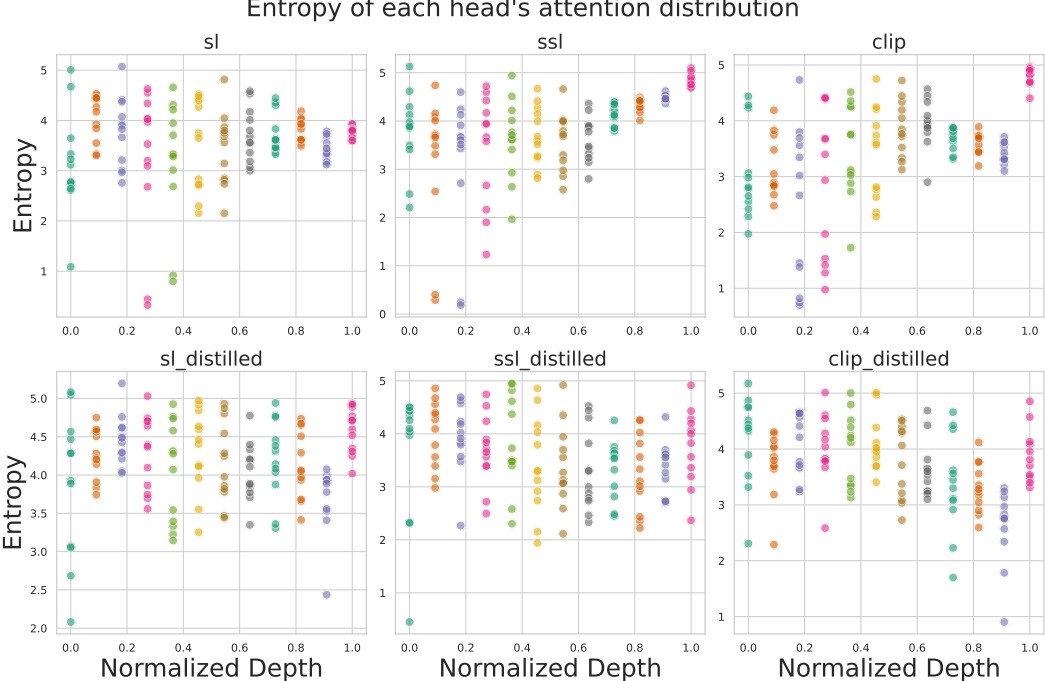

Figure 10: The distribution of attention entropy across different heads varies among models. Pre-trained models tend to exhibit homogeneous and elevated levels of attention entropy in higher layers, indicating that different heads are focusing on similar information. In contrast, Masked Distillation effectively diversifies the attention across various heads.

the target. Concurrently, we observed that the use of a tokenizer induces a preference for low-frequency features, which may be attributed to the inherent tendency of discrete tokens to discard finer details (Vahdat et al., 2018).

## C  OOD EXPERIMENTS

In Figure 11, we present the evaluation of the model's shape-texture bias. Our research suggests that Masked Distillation does not consistently enhance the model's shape bias. Pretrained models inherently possess a strong shape bias, a trait that is also observed in human decision-making. Fine-tuning amplifies the model's texture bias, but Masked Distillation mitigates this tendency to some extent.

We further argue that the claim that Masked Distillation leads to more human-like decision-making is not rigorously substantiated. To demonstrate the underlying reasons for this, following Geirhos et al. (2021), we evaluated the model's decision-making tendencies on 17 out-of-distribution (OOD) datasets, with the aggregated results displayed in Figure 12. Figure 12a demonstrates that Masked Distillation improves the performance in OOD settings. However, when considering the impact of accuracy on decision-making tendencies, it is worth noting that a model with a 95% accuracy has at most a 5% probability of making an error. Consequently, the probability of alignment with human decision-making would naturally be higher for a model with higher accuracy. Following Geirhos et al. (2021), we conducted normalization since our destination is to evaluate error consistency —that is, the congruence between human and model errors—rather than accuracy consistency. We found that Masked Distillation does not make the model decisions more human-like.

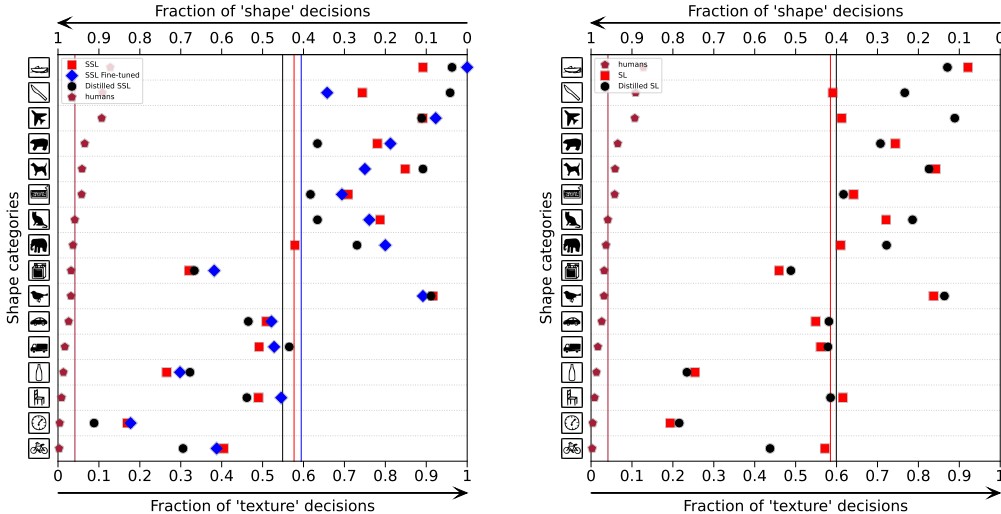

Figure 11: Analysis of shape bias under the teacher supervision in DINO ViT-B/16 (**left**) and DeiT ViT-B/16 (**right**).

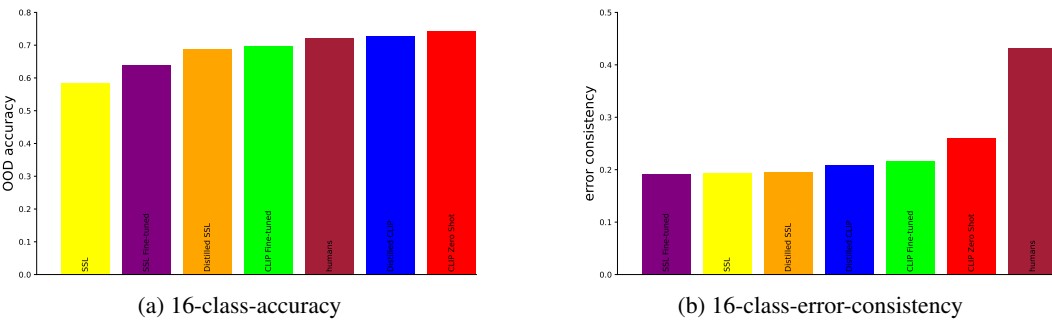

(a) 16-class-accuracy

(b) 16-class-error-consistency

Figure 12: Evaluation of the model's decision-making tendencies on 17 out-of-distribution (OOD) datasets.

## D    HYPERPARAMETERS

We provide a list of training hyperparameters. The hyperparameters for Masked Distillation are delineated in Table 1. The training hyperparameters for the baseline that we propose is the column **CLIP/ViT-B/16**. By aligning these hyperparameters, our results can be easily reproduced.

Table 1: Hyperparameters for Different Models on ImageNet-1K

| Hyperparameters | VQ-KD | BEIT v2/Base Size | CLIP/ViT-B/16 |
|---|---|---|---|
| Encoder Layers | 12 | 12 | - |
| Decoder Layers | 1 | - | - |
| Hidden Size | 768 | 768 | - |
| FFN Inner Hidden Size | 3072 | 3072 | - |
| Attention Heads | 12 | 12 | - |
| Layer Scale | - | 0.1 | - |
| Patch Size | $16 \times 16$ | $16 \times 16$ | - |
| Codebook Size | $8192 \times 32$ | - | - |
| Relative Positional Embeddings | - | ✓ | ✓ |
| Shared Relative Positional Embeddings | - | ✓ | ✗ |
| Training Epochs | 100 | 300 | 50 |
| Batch Size | 512 | 2048 | 1024 |
| Adam $\beta$ | $(0.9, 0.99)$ | $(0.98, 0.999)$ | $(0.9, 0.95)$ |
| Peak Learning Rate | $2 \times 10^{-4}$ | $1.5 \times 10^{-3}$ | $5 \times 10^{-5}$ |
| Minimal Learning Rate | $1 \times 10^{-5}$ | $1 \times 10^{-5}$ | $1 \times 10^{-7}$ |
| Learning Rate Schedule | Cosine | Cosine | Cosine |
| Warmup Epochs | 5 | 10 | 10 |
| Layerwise Learning Rate Decay | - | - | 0.6 |
| Weight Decay | $1 \times 10^{-4}$ | 0.05 | 0.05 |
| Gradient Clipping | ✗ | 3.0 | ✗ |
| Dropout | ✗ | ✗ | ✗ |
| Stoch Depth | - | 0 | 0 |
| Label Smoothing | - | - | 0.1 |
| Resize & Crop | RandomResizeAndCrop | RandomResizeAndCrop | - |
| Color Jitter | 0 | 0.4 | 0.4 |
| RandomHorizontalFlip | 0.5 | 0.5 | - |
| Input Resolution | $224 \times 224$ | $224 \times 224$ | $224 \times 224$ |
| Min Crop Scale | 0.08 | 0.2 | 0.08 |
| AutoAug | - | - | rand-m9-mstd0.5-inc1 |
| Random Erase prob (reprob) | - | - | 0.25 |
| Random Erase mode (remode) | - | - | pixel |
| Random Erase count (recount) | - | - | 1 |
| Mixup | - | - | ✗ |
| Cutmix | - | - | ✗ |
| Freeze Patch Projection | - | - | ✓ |

