# OpenReview forum: "Towards Understanding Masked Distillation"
_ICLR.cc/2024/Conference — Submitted to ICLR 2024_

### Official Review · Reviewer_xu9z · 2023-10-29

**Soundness:** 2 fair
**Presentation:** 3 good
**Contribution:** 2 fair
**Rating:** 5
**Confidence:** 4

**Summary:**

This paper delves into the intricacies of Masked Distillation (MD), a specific variant of masked image modeling, and scrutinizes its inductive biases. The paper is heavily empirical and makes several key observations:
1. MD mitigates attention homogenization, where the model's response is diffused across a broad and unrelated area for specific queries; 2. the features learned through MD exhibit lower frequency than those learned without MD;
3. MD produces higher singular values for each hidden layer's activations;
4. the entropy of the features learned through MD is higher;
5. MD provides a smoother loss landscape for training.

**Strengths:**

1. Despite the paper's structure needing significant improvement, it is relatively straightforward to comprehend the various analyses.
2. The exploration of how MD shapes the model's learned representations is crucial.
3. The authors provide detailed experimental procedures, enhancing reproducibility.

**Weaknesses:**

1. The paper lacks a comparison of the downstream performance of models trained with and without MD. While the authors demonstrate how the features differ, the impact of these differences on downstream performance remains unclear.
2. The empirical results' distinction from previous analyses on regular MIM (as presented in 'Revealing the Dark Secrets of Masked Image Modeling') is unclear. Given the similarities between MIM and MD, I worry that the work, which shows MD performing similarly to MIM, may have limited impact.
3. Several analyses (attention distance, entropy) appear to be borrowed from the aforementioned previous work.
4. The introduction lacks structure. It disproportionately focuses on MIM, whereas I would expect a more detailed introduction and comparison between MIM and MD for a paper centered on MD.

**Questions:**

1. In Figure 10, the authors state that MD diversifies attention across various heads, but does a smaller entropy range necessarily imply similar attentions? Could visualizing the distance between different attention heads provide more clarity?
2. The authors claim that MD amplifies the model’s preference for low-frequency features while enhancing the preference for high-frequency features in the lower layers. However, from my understanding of the figure, it seems that MD only promotes high-frequency features in the lower layers for SSL (blue). Could you clarify?
3. I don't see the necessity of splitting section 3 and section 4 into two sections. Both sections compare different properties between MD and non-MD.

---

### Official Review · Reviewer_2wDN · 2023-10-31

**Soundness:** 3 good
**Presentation:** 3 good
**Contribution:** 2 fair
**Rating:** 3
**Confidence:** 4

**Summary:**

In this paper, the authors make a comprehensive understanding of masked distillation. This understanding includes some properties inherited from masked image modeling, and some unique properties of masked distillation. These investigations explain the behaviors under the strong performance of mask distillation models such as attention homogenization, representation folding and smooth loss landscapes.

**Strengths:**

1. The viewpoint of this paper is interesting.
2. Some analysis of current technicals is valuable, such as the modality differences, and the gap between pre-training and downstream tasks.
3. The unique feature of mask distillation is proposed that inspires modern training pipeline design.

**Weaknesses:**

1. In figure 2,3,4, some confused tags 'sl', 'ssl', 'clip' are not explained. These tags are not consist with the analysis in the paper with 'DINO', 'CLIP', 'supervised models' mentioned.
2. This paper gives some new point of masked distillation and explains how it works. However, it remain unclear that how to design a better pipeline with these findings.

**Questions:**

1. Some legends without explanation make me confused on some details in the paper. It will be better that the authors fix it and re-review again for me.

---

### Official Review · Reviewer_65e1 · 2023-11-01

**Soundness:** 1 poor
**Presentation:** 1 poor
**Contribution:** 1 poor
**Rating:** 1
**Confidence:** 3

**Summary:**

This paper delves into the efficacy of masked distillation as a self-supervised learning approach for vision tasks, highlighting four key findings:

"Masked Distillation imparts the model with intrinsic mutual information attributes."

"Masked Distillation amplifies the expressive potential of the attention module."

"Masked Distillation alleviates the issue of representation folding in higher layers."

"Masked Distillation contributes to the smoothing of the loss landscape."

Additionally, the paper delves into the discussion of "model design and decision preferences."

**Strengths:**

Masked Distillation seems to be a promising self-supervised learning method that is not understood well. Investigation of this is an interesting research direction.

**Weaknesses:**

1. The paper often uses unscientific and unexplained language
e.g. "We propose a natural hypothesis: Masked Distillation imbues the model with attributes inherent to MI". It's not clear how masked distillation imbues the model with them and how the experiments designed are helpful in validating this.

"Masked Distillation inherits the inductive bias of attention from MIM": "inductive bias of attention", to the best of my knowledge, is not a well-defined term. Furthermore, the paragraph following this is full of sentences such as "This phenomenon may be attributed to their extreme homogenization of background responses, suggesting that the degree of locality and diversity attained is contingent upon the original model" which are not backed with evidence (or at least no evidence is referred to).

2. No explanation is given for why the properties above help Masked Distillation. The connection of these claims e.g. "prevents overfitting" are not explained / proven using evidence.

3. Explanation of experiments is hugely lacking.
The paper attempts to explain Masked Distillation through exploratory experiments, but the exact details of these experiments are not presented in the paper clearly. Moreover,

**Questions:**

1. What exactly is the new analytical framework the authors mentioned they had created?

---

### Official Review · Reviewer_hmEG · 2023-11-02

**Soundness:** 1 poor
**Presentation:** 3 good
**Contribution:** 1 poor
**Rating:** 3
**Confidence:** 5

**Summary:**

The paper analyzes masked distillation methods, which use a teacher model as a MIM target instead of raw pixels. The experiments are designed similarly to [A]. it shows that the model trained with masked distillation is different from SL, SSL, and CLIP-trained models, like the MIM-trained model.

[A] WHAT DO SELF-SUPERVISED VISION TRANSFORMERS LEARN?, ICLR 2023

**Strengths:**

- The paper includes diverse analysis for masked distillation.

**Weaknesses:**

- I think the masked distillation is an improved version of MIM. Thus, it should be compared with MIM. However, the paper's experiments compare masked distillation with SL, SSL, and CLIP teacher models. The results show that masked distillation is similar to MIM. I'm hard to find the value of this experiment. I think masked distillation is naturally similar to MIM, and verifying it has a minimum contribution.

- Experiment metrics and messages are significantly similar to [A]. The following metrics overlap with [A]: attention distance, frequency spectrum,  relative singular value (token-level, image-level), and normalized MI. This paper can be considered as a masked distillation version of [A], which significantly limits the novelty.
  - [A] WHAT DO SELF-SUPERVISED VISION TRANSFORMERS LEARN?, ICLR 2023

**Questions:**

- "Masked distillation" is used to represent many methods [B, C, D, E]. I recommend authors add a clear definition of masked distillation at the beginning of the paper.
  - [B] Masked Generative Distillation, ECCV 2022
  - [C] MASKED DISTILLATION WITH RECEPTIVE TOKENS, ICLR 2023
  - [D] Supervised Masked Knowledge Distillation for Few-Shot Transformers, CVPR 2023
  - [E] MaskedKD: Efficient Distillation of Vision Transformers with Masked Images, arxiv

- I think the paper should compare masked distillation with MIM for diverse variants. Comparing masked distillation with SL, SSL is similar to [A]. Thus, it is hard to contribute to other researchers.

- What masked distillation does the paper use for experiments? As described in Fig. 3 (c), there are diverse variants of masked distillation. Can the paper's implementation (*_MD) be enough to represent all variants of masked distillation?

- I believe analysis metrics are strongly related to ImageNet-1k performance. What is the ImageNet-1k performance of SL, SSL, CLIP, SL_MD, SSL_MD, and CLIP_MD?

- [F] is a pioneer work in masked distillation research and also strongly related to this paper. I recommend authors cite [F] and add discussion with it.
  - [F] Exploring Target Representations for Masked Autoencoders, arxiv

- Attention diversity and homogeneity can't be measured by entropy. They should be measured by the distance between probabilities, such as KL divergence or cross-entropy. Please use the correct metrics for them.

- The main quantitative results for attention diversity and homogeneity (Figure 10) are in the appendix. I think the main results should be included in the main script. The current paper is the same with no quantitative results for attention diversity and homogeneity.

- What is the meaning of frequency analysis in transformer architecture? I believe frequency is reasonable for ConvNet but meaningless for Transformer.

- In Fig 6 (a), SL_MD has lower NMIs compared to SL, which is different from the explanation of the paper.

- How does the loss landscape for SL and SL_MD look like? The loss landscape is a good tool for measuring over-fitting. Thus, I want to know why SL is excluded from the experiment.

- Fu et al. 2023 [G] is a study on SGD. But, most of ViT uses AdamW optimizer. Can the relation between hessian and optimization stability be generalized to AdamW optimizer?
  - [G] When and why momentum accelerates sgd:an empirical study, 2023

---

### Meta-Review · Area_Chair_BWNS · 2023-12-01

**Metareview:**

Reviewers raised several concerns about the paper, primarily around significant lack of clarity in exposition and experimental setup. There was no response to these. We therefore uphold the reviewers' consensus, and do not recommend the paper for publication at present.

**Justification For Why Not Higher Score:**

Unanimous Reject recommendation from reviewers, with no author response.

**Justification For Why Not Lower Score:**

N/A

---

### Decision · Program_Chairs · 2024-01-16

Reject